

# Living upside down: patterns of red coral settlement in a cave

Federica Costantini[1,2,3], Luca Rugiu[4], Carlo Cerrano[3,5] and Marco Abbiati[2,3,6,7]

[1] Department of Biological, Geological and Environmental Sciences (BiGEA), University of Bologna, Ravenna, Italy
[2] Centro Interdipartimentale di Ricerca per le Scienze Ambientali (CIRSA), Ravenna, Italy
[3] CoNISMa, Roma, Italy
[4] Section of Ecology, Department of Biology, University of Turku, Turku, Finland
[5] Department of Life and Environmental Science (DiSVA), Università Politecnica delle Marche, Ancona, Italy
[6] Istituto di Scienze Marine (ISMAR), Consiglio Nazionale delle Ricerche, Bologna, Italy
[7] Department of Cultural Heritage, University of Bologna, Ravenna, Italy

Corresponding author
Federica Costantini,
federica.costantini@unibo.it

## ABSTRACT

**Background.** Larval settlement and intra-specific interactions during the recruitment phase are crucial in determining the distribution and density of sessile marine populations. Marine caves are confined and stable habitats. As such, they provide a natural laboratory to study the settlement and recruitment processes in sessile invertebrates, including the valuable Mediterranean red coral *Corallium rubrum*. In the present study, the spatial and temporal variability of red coral settlers in an underwater cave was investigated by demographic and genetic approaches.

**Methods.** Sixteen PVC tiles were positioned on the walls and ceiling of the Colombara Cave, Ligurian Sea, and recovered after twenty months. A total of 372 individuals of red coral belonging to two different reproductive events were recorded. Basal diameter, height, and number of polyps were measured, and seven microsatellites loci were used to evaluate the genetic relationships among individuals and the genetic structure.

**Results.** Significant differences in the colonization rate were observed both between the two temporal cohorts and between ceiling and walls. No genetic structuring was observed between cohorts. Overall, high levels of relatedness among individuals were found.

**Conclusion.** The results show that *C. rubrum* individuals on tiles are highly related at very small spatial scales, suggesting that nearby recruits are likely to be sibs. Self-recruitment and the synchronous settlement of clouds of larvae could be possible explanations for the observed pattern.

## INTRODUCTION

Recovery and resilience of sessile benthic organisms mostly depend on their early life history stages such as dispersal, settlement and recruitment (*Hughes et al., 2000*; *Pineda et al., 2007*). It has been shown that abiotic (*Torrents & Garrabou, 2011*) and biotic (*Lindsay, Wethey & Woodin, 1997*) factors, as well as larval behaviour (*Martínez-Quintana et al., 2014*), may influence dispersal and larval mortality during the pre-settlement period.

Settlement is influenced by events occurring during the planktonic stage (*Babcock & Mundy, 1996*). After settlement, other sources of mortality (e.g., intraspecific competition, predation, detachment from the substrate) can affect the recruitment process (*Perkol-Finkel et al., 2008*; *Santangelo et al., 2012*). Larval dispersal and recruitment play a primary role in maintaining genetic diversity, but these processes are stochastic, and may contribute to a chaotic genetic patchiness (*Johnson & Black, 1982*). These effects are more evident at fine spatial scales below the expected range of larval dispersal of the species (*Eldon et al., 2016*). In marine invertebrates chaotic genetic patchiness seems related mainly to high variance in reproductive success (*Hedgecock, 1994*), collective dispersal (*Broquet & Yearsley, 2012*) and asynchronous local population dynamics (*Eldon et al., 2016*).

Early life stages (from larval release to recruitment) in marine invertebrates such as sponges, ascidians and cnidarians, have been investigated using different tools such as laboratory experiments on larval behaviour (*Guizien et al., 2012*; *Martínez-Quintana et al., 2014*), on settlement and metamorphosis on different substrates (*Bavestrello et al., 2000*), field experiments on settlement and post-settlement processes (*Fraschetti et al., 2002*), mathematical simulation by biophysical circulation modelling (*Guizien et al., 2006*), and empirical evidences from population genetics (*Hedgecock, Barber & Edmands, 2007*; *Eldon et al., 2016*). Moreover, recruitment rates, and their variability in space and time, can be estimated directly, using settlement tiles (*Bramanti et al., 2007*; *Green & Edmunds, 2011*; *Santangelo et al., 2012*; *Bramanti & Edmunds, 2016*); while spatial genetic structure (SGS; e.g., genetic variability, relatedness) can provide indirect estimates of recruitment patterns and variability (*Brazeau, Sammarco & Atchison, 2011*; *Smilansky & Lasker, 2014*). The strong SGS observed in Anthozoa suggest that recruitment is often local, probably as a result of the short effective dispersal of larvae (*Costantini, Fauvelot & Abbiati, 2007*; *Ledoux et al., 2010a*). Recruitment coming only or mostly from local populations (self-recruitment) can lead to an impoverishment of the genetic variability and thus decreasing the population's resilience to stressors (*Brazeau, Sammarco & Atchison, 2011*; *Lasker, 2013*) but could also enhance population survival through local adaptation (*Sanford & Kelly, 2011*). However, to date, only few studies have analysed the SGS in coral settlers and recruits (*Brazeau, Sammarco & Atchison, 2011*; *Torda et al., 2013*; *Smilansky & Lasker, 2014*).

Underwater caves (sensu *Rastorgueff et al., 2015*) represent a naturally fragmented and confined habitat not exposed to the strong currents, which often provide a natural protection from disturbances associated with waterborne substances (*Garrabou & Harmelin, 2002*). Due to their high species richness, caves are considered a Mediterranean biodiversity reservoir (*Gerovasileiou & Voultsiadou, 2012*). Underwater caves represent, therefore, an excellent natural mesocosm to investigate the recruitment processes without adding other stochastic external disturbances. Moreover, the understanding of recruitment processes in cave species assemblages will be pivotal to forecast their ability to recover after disturbances, and to understand if they can act as refugia for populations living outside the caves.

The red coral (*Corallium rubrum* L. 1758) is one of the abundant species inhabiting Mediterranean caves due to its preference for dim-light conditions and downwards-facing surfaces (*Laborel & Vacelet, 1961*; *Garrabou & Harmelin, 2002*; *Virgilio, Airoldi & Abbiati, 2006*). Red coral is a gonocohoric species with internal fertilization. Gonadal

development follows an annual cycle with a synchronized release in summer(*Santangelo et al., 2003*). Planulae are internally brooded and released once a year over a period of approximately two weeks between the end of July and early August (*Santangelo et al., 2003*; *Bramanti et al., 2005*). *C. rubrum* populations are threatened by several stressors: smothering by sediments, harvesting (*Tsounis et al., 2013*), climate change (*Bramanti et al., 2013*; *Cerrano et al., 2013*), and heat waves (*Cerrano, Bavestrello & Bianchi, 2000*). Several recruitment studies using settlement tiles have been carried out on shallow-water red coral populations inhabiting vertical cliffs or small crevices (*Bramanti, Magagnini & Santangelo, 2003*; *Bramanti et al., 2007*; *Santangelo et al., 2012*). High variability in the density of recruitment between different sites has been found, and has been attributed to biological interactions (e.g., competition for space, predation, overgrowth) (*Bramanti, Magagnini & Santangelo, 2003*; *Santangelo et al., 2012*). Population genetic studies, conducted only on adult colonies, have shown large heterozygosity deficiencies, suggesting the occurrence of inbreeding (non-random mating) within populations (*Costantini, Fauvelot & Abbiati, 2007*; *Ledoux et al., 2010a*; *Ledoux et al., 2010b*; *Aurelle et al., 2011*) and a strong genetic structure at distances of less than one meter (*Costantini, Fauvelot & Abbiati, 2007*; *Ledoux et al., 2010a*).

In the present study, we investigated two cohorts of red coral recruits on settlement tiles deployed in a Mediterranean submarine cave acting as an experimental mesocosm. Specifically, we have analysed the variability of biometric parameters of the two cohorts of settlers (e.g., abundance of settlers, diameter, height, number of polyps). By means of microsatellite loci, the relatedness and the fine spatial genetic structure within and between the two cohorts have been analysed in order to provide additional information on early life characteristics and population dynamics of this species. Understanding red coral recruitment processes is of key importance to unveil the drivers of the population dynamics of the species and hence the potential recovery of the overexploited/threatened populations.

## MATERIALS AND METHODS

### Study area and experimental design

The study was conducted inside the Colombara (or Marcante) cave (Lat 44°18′35″N, Long 9°10′37″E) located on the east coast of the Gulf of San Fruttuoso (Italy). The cave, 10 m long, stands from 34 to 39 m depth on a rocky cliff southward oriented. The cave walls host a rich assemblage of sessile invertebrates typical of the sublittoral caves of the northwest Mediterranean Sea (*Morri et al., 1986*) with many sponges, corals, bryozoans, polychaetes, and tunicates.

Field experiments were approved by the Marine Protected Area of Portofino and by the Università Politecnica delle Marche (Authorization n. 3/2011 (n. prot. 449/2-1-5.) and authorization no 4/2012 (Protoc. No 409/2-1-1)).

In June 2010, about three weeks before the start of red coral spawning (*Santangelo et al., 2003*), sixteen 20 × 20 cm white PVC tiles, drilled in the centre, were fixed inside the cave by steel screws. PVC was selected owing to the success in previous experiment on larval recruitment (C Cerrano, pers. comm., 2010; but see also *Kennedy et al., 2017*). Moreover, having positive buoyancy, the risk of detachment from the ceiling was avoided.

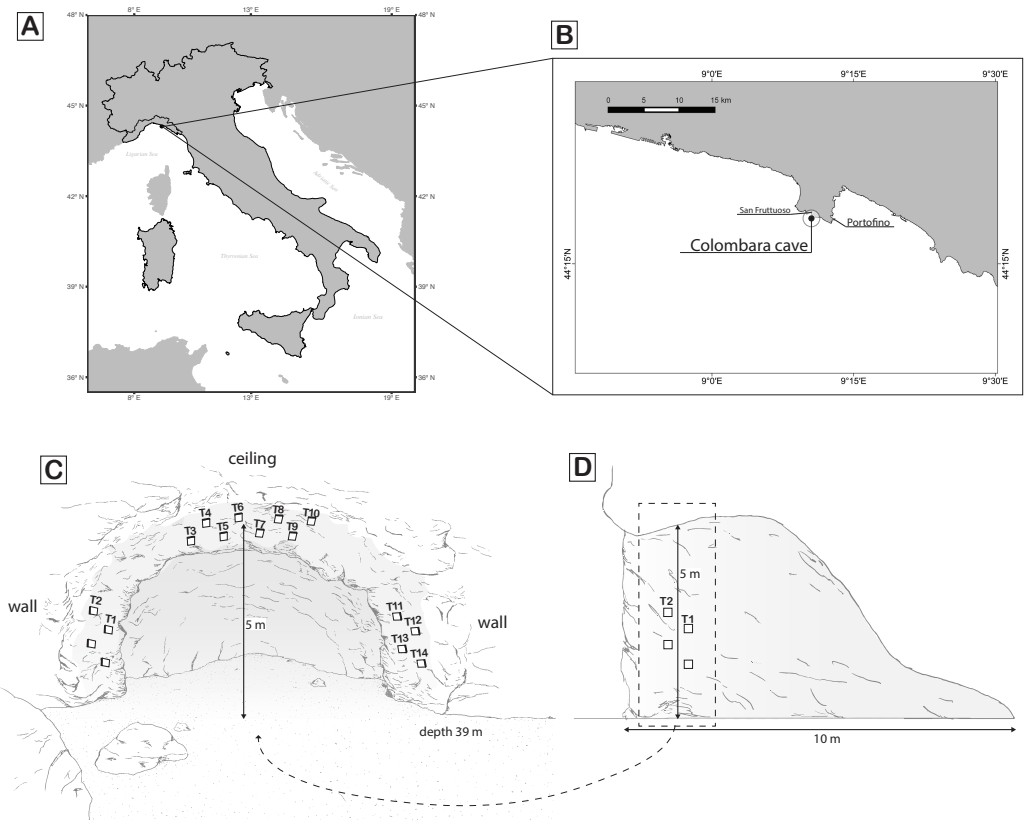

**Figure 1** **Maps of sampling location and scheme of the sampling design.** (A) Overview of the geographic location of the Colombara cave; (B) zoom of the Portofino promontory where the cave is located; (C) front of the cave and scheme of the tiles deployed; (D) profile of the cave. White rectangles represent the tiles. Rectangles without number represent the lost tiles. Drawing made by Mancuso FP.

In order to test if recruitment is affected by orientation, one plot of four tiles was located on the right vertical wall, another plot on the left vertical wall, and two plots on the ceiling of the cave (Fig. 1). Each tile was attached to the rock, 1 m from the entrance and at a minimum distance of 30 cm from any another tiles, to avoid possible mutual interference. The red coral population distribution into the cave is patchy, showing an average density of $349 \pm 215$ col/m$^2$ (*Cattaneo-Vietti, Bavestrello & Senes, 1993*). Tiles were attached in low-density areas of the red coral population trying to limit as much as possible the breakage of surrounding colonies.

In February 2012, after two reproductive events (summer 2010 and summer 2011), the tiles were removed ($n = 14$ as two located on one vertical wall were lost, Fig. 1). Recovered tiles were fixed in 80% ethanol and transferred to the laboratory. A picture of each tile was taken with a NIKON camera on a stereomicroscope NIKON SMZ 1500 and analysed with IMAGE J software version 1.24o (*Schneider, Rasband & Eliceiri, 2012*) to study the spatial distribution of the red coral individuals. The position of each individual was marked and size (diameter) was measured by averaging the minimum and maximum width. All the

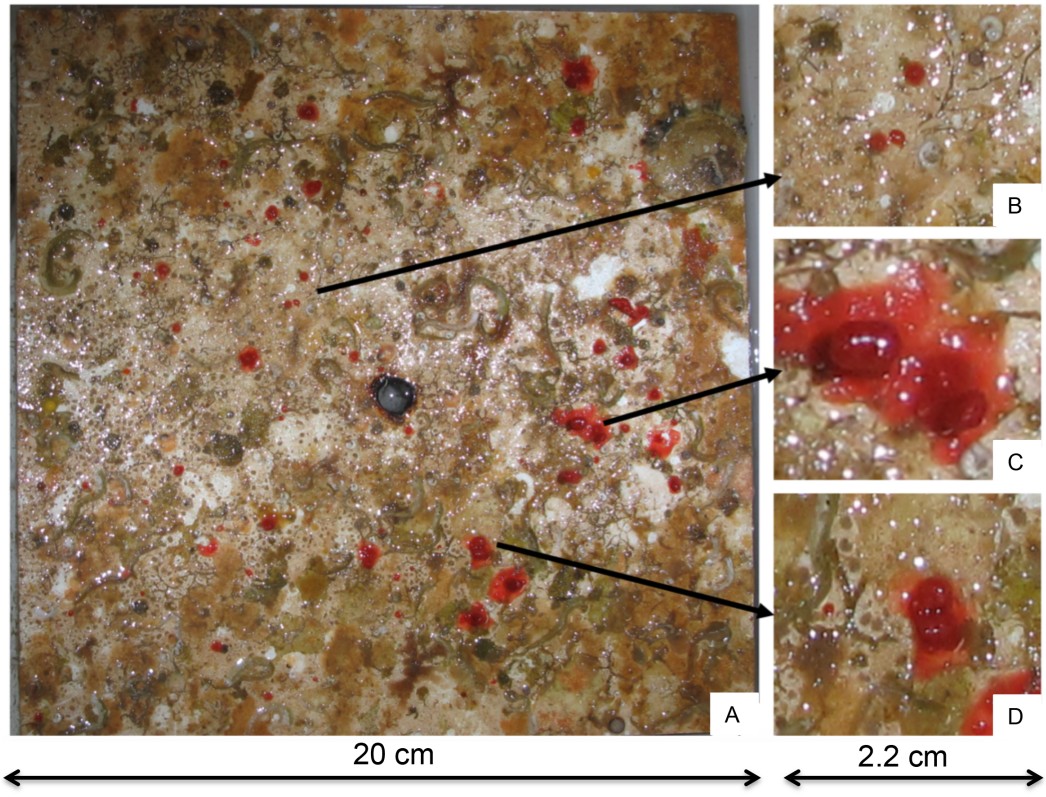

**Figure 2** **Example of a PVC tile recovered from the Colombara cave after two years.** (A) Tile T7, (B) zoom of a recruit, (C) a juvenile, (D) a recruit probably derived by two merged planulae.

individuals were then removed from the tiles and for each of them; the number of polyps was counted under the dissecting microscope. Polyps were removed from each individual and stored in plastic tubes with 80% ethanol at 4 °C for the subsequent DNA extraction.

Based on the literature on red coral early life stages (*Bramanti et al., 2005*) the age of each individual was estimated on the basis of its height: individuals with an encrusting button shape and colonies with height equal to zero were assigned to the cohort 2011 (hereafter recruits); individuals that developed in height forming a small branch were assigned to the cohort 2010 (hereafter juveniles) (Fig. 2).

To check if a sort of cave-effect affected the pattern of recruitment, in the same period an additional series of plots with four PVC tiles were deployed on a vertical cliff out of the cave (Punta del Faro) where the red coral population has the same range of density as that of the cave ($425 \pm 100$ colonies $\times$ m$^2$) (*Bavestrello et al., 2015*). Plots were fixed at 35, 45, 55 and 70 m depth.

## Measuring growth performances

The size structure of red coral individuals was obtained by analysing the frequency distributions of the basal diameter (for both recruits and juveniles) and of colony height (only for juveniles). The correlations between the parameters (diameter vs. height; diameter

vs. number of polyps; height vs. number of polyps) were analysed by Pearson's coefficient. Moreover, we tested the temporal and spatial variations of the abundance of individuals by two separate one-way ANOVAs based on 9,999 permutations with Cohort (two levels; fixed; recruits and juveniles) and Position (two levels; fixed; ceiling and walls) as factors using PRIMER v6 software program (*Clarke & Gorley, 2006*).

## Measuring genetic variability and structuring

Due to the small number of *C. rubrum* individuals found on the tiles deployed on the walls, the molecular analysis was carried out only on the individuals occurring on the tiles deployed on the ceiling of the cave.

Total genomic DNA was extracted from each individual (1 to 4 polyps per colony) following the CTAB protocol and purified by standard chloroform procedure. Seven microsatellite loci COR9, COR46, L7, COR58, MIC20, MIC24, MIC26 (*Costantini & Abbiati, 2006*; *Ledoux et al., 2010b*) were amplified either as single locus using the protocol by *Costantini & Abbiati (2006)* and by *Ledoux et al. (2010b)* or in multiplex using a QIAGEN Multiplex PCR Kit. Genotyping was performed by MACROGEN INC. Allele sizing was determined with Peak Scanner v1.0 software. Genotypic linkage disequilibrium at each pair of loci for each was tested using FSTAT v.2.9.3.2 (*Goudet, 2001*). Significance of each pairwise comparison was tested using 3,360 permutations of the data.

Scoring errors due to stuttering, large allele dropout and null alleles were controlled with MICROCHECKER version 2.2.3 (*Van Oosterhout et al., 2004*). Estimated frequencies of putative null alleles were subsequently calculated for each locus using the expectation maximization (EM) algorithm of *Dempster, Laird & Rubin (1977)* implemented in FREENA (*Chapuis & Estoup, 2007*). Red coral settlers sharing the same multilocus genotype (MLG) were identified using GENALEX version 6.1 (*Peakall & Smouse, 2006*). Identical MLGs can result from two distinct sexual reproduction events. To test this, the unbiased probability of identity ($P_{ID}$) that two sampled individuals shared identical MLG by chance (*Kendall & Stewart, 1977*) was computed. The total number of alleles ($N_a$), observed ($H_o$), and unbiased expected ($H_e$) heterozygosities (*Nei, 1973*) were calculated for each locus and for each cohort of settlers (either recruits and juveniles) using GENETIX software package version 4.03 (*Belkhir et al., 2000*). Allelic richness (Ar) was calculated after controlling for differences in sample size using a rarefaction approach implemented in HP rare with a common sample size of 54 settlers (*Kalinowski, 2005*). The $f$ estimator of $F_{IS}$ (*Weir & Cockerham, 1984*) was computed, and significant departures from the Hardy–Weinberg equilibrium were tested using Fisher's exact test in GENEPOP version 3.4 (http://genepop.curtin.edu.au/; *Raymond & Rousset, 1995*), with the level of significance determined by a Markov chain randomization. Significant differences in genetic diversity ($H_o$, $H_e$ and Ar) between recruits and juveniles were tested using a Student's $t$-test. For all the analyses, when necessary, a correction for multiple tests was applied with a false discovery rate of 0.05 (*Benjamini & Hochberg, 1995*).

To determine whether individuals were more related than expected under panmixia, the $R_{XY}$ pairwise relatedness coefficient (*Queller & Goodnight, 1989*) was computed separately for recruits and for juveniles. This index varies between 0 and 0.5 with $R_{XY} = 0.5$ for full-sib

relationships, $R_{XY} = 0.25$ for half-sibs and $R_{XY} = 0$ for unrelated individuals in an infinitely large panmictic population (*Peakall & Smouse, 2006*). The observed mean and variance of $R_{XY}$ were compared with their expected distribution under the null hypothesis of panmixia using 1,000 permutations of alleles as implemented in IDENTIX (*Belkhir, Castric & Bonhomme, 2002*). The null distribution was obtained by a conventional Monte Carlo resampling procedure, which randomly selected 10,000 genotypes without replacement and then recalculating the statistic. Null hypothesis could be rejected with a significance level of 5%, given that the observed value of the statistic was above the 95% level of the resampled statistics.

An exclusion test, performed in GENECLASS 2.0 (*Piry et al., 2004*), was used to test whether individuals found in one tile was more similar to the individuals settled on the same tile. First, the likelihood that an individual belonged to a particular tile was computed with a Bayesian criterion of *Rannala & Mountain (1997)*. Then, this likelihood was compared to a distribution of likelihoods of 10,000 genotypes simulated from each candidate tile with a Monte Carlo algorithm. An individual was excluded from its tile when the probability of exclusion was greater than 95% ($P$ or $\alpha \leq 0.05$). The second part of this Bayesian analysis utilized the probabilities that the individuals excluded from their sampling tile originated from one of the other tiles. Thus, individuals that were excluded from their sampling tile when $P \leq 0.05$, were assigned to another tile when $P \geq 0.1$.

The value of effective population size ($Ne$) for each cohort was inferred using the standard linkage disequilibrium method with *Waples (2006)* correction. The computations were performed with LDNe under the random-mating model, excluding rare alleles with frequencies of less than 0.02 and using the Jackknife option to estimate confidence intervals (*Waples & Do, 2008*; *Waples & Do, 2010*).

To account for unbalanced sample sizes between the recruits and juveniles, genotypic differentiation was tested with an exact test (Markov chain parameters: 1,000 dememorizations, followed by 1,000 batches of 1,000 iterations per batch), and the $P$ value of the log-likelihood (G)-based exact test (*Goudet et al., 1996*) was estimated in GENEPOP.

A Bayesian approach implemented in the program STRUCTURE version 2.2 (*Pritchard, Matthew & Donnelly, 2000*; *Falush, Stephens & Pritchard, 2007*) was used to estimate the number of genetic clusters, K, within the entire sample (i.e., recruits + juveniles). Mean and variance of log likelihoods of the number of clusters for $K = 1$ to $K = 10$ were inferred by running structure ten times with 1,000,000 repetitions each (burn-in = 100,000 iterations) under the admixture ancestry model and the assumption of correlated allele frequencies among samples. Due to the presence of null alleles, the clustering analysis was conducted on the original data set, using the option of null alleles coded as recessive alleles described in *Falush, Stephens & Pritchard (2007)*. Mean likelihoods of K from ten runs were plotted using STRUCTURE HARVESTER 0.56.3 (available at http://taylor0.biology.ucla.edu/struct_harvest/). Results of all runs were averaged in CLUMPAK server (*Kopelman et al., 2015*). Moreover, since the Structure analysis could be inflated by the HW disequilibrium, a discriminant analysis of principal components (DAPC), available in the Adegenet package (*Jombart, Devillard & Balloux, 2010*) for R (*R Development Core Team, 2012*), was also performed. This technique is designed for

multivariate genetic data (multi-locus genotypes). It maximizes the variation between groups by first performing a principal component analysis (PCA) on pre-defined groups or populations and then uses the PCA factors as variables for a discriminant analysis (DA), thus ensuring their independence.

Spatial autocorrelation analyses among individuals were performed with SPAGEDI (*Hardy & Vekemans, 2002*). The Loiselle's kinship coefficient ($\phi$ij; *Loiselle et al., 1995*) was used since it is not dependent on Hardy–Weinberg (HW) equilibrium conditions (*Hardy, 1999*) and it has been proven to be very effective in determining spatial genetic structure (*Vekemans & Hardy, 2004*). Two different analyses were performed. To test the spatial autocorrelation within the cave, distance categories were set based on the distance between tiles and considering the distance between two individuals found within the same tile as zero. Then, spatial autocorrelation analyses within those tiles containing more individuals (T4, T5, T7, T8, T9 and T10; see results) were carried out. Taking into account the distance between individuals, the distance categories were set in such a way that the number of pairwise comparisons within each distance category was approximately constant. Statistical significance was based on permuting individual locations among all individuals 10,000 times and calculating upper and lower 95% confidence interval for each distance class.

## RESULTS

### Red coral recruitment

In the plots positioned into the cave, 372 individuals were observed on the 14 tiles. *Corallium rubrum* settled on every tile deployed on the ceiling of the cave, but recruitment failed on three out of eight tiles on the walls (T1, T13, T14). In fact, of the 372 individuals, 350 were found on the ceiling tiles, and only 22 on the tiles deployed on the walls, corresponding to a significantly higher density on the ceiling than on walls (ANOVA: $F_{1,14} = 10.78$; $P < 0.01$, Fig. 3). Of the 350 individuals on the ceiling, 278 were recruits and 72 were juveniles, corresponding to a density of $34.75 \pm 23.86$ /400 cm$^2$ and of $9 \pm 6.82$ / 400 cm$^2$, respectively (ANOVA: $F_{1,26} = 5.99$; $P < 0.05$, Fig. 2). The 22 individuals found on the wall were all recruits (Fig. 3). On the tiles positioned along the vertical cliff, at all depths no recruits were found when checked in summer 2011.

### Growth performances

The size/recruit structure showed a monotonic and decreasing pattern, in which recruits in the first class (recruits with diameter < 1.5 mm) represented the dominant class (Fig. 4A). Only three recruits (1%) had a diameter >6 mm, and they might have been formed by merging of two or more planulae (Fig. 2) as observed by C Cerrano (pers. comm., 2010). The number of polyps per recruit ranged from 1 to 22, with 70.66% of the recruits presenting 1–2 polyps, 22% between 3–4 polyps and 7.33% more than four polyps (mean $\pm$ SD = $2.3 \pm 2.1$ polyps/recruit). Pearson's coefficient showed a low correlation between these two variables ($r = 0.4$) (Fig. 4C).

The size structure of juveniles showed a more variable trend, with diameter values ranging from 0.15 mm to 8.95 mm with an average value of $2.5 \pm 1.5$ mm (Fig. 5A). The number of polyps ranged from 0 to 22 (average number of $9.21 \pm 4.6$) (Fig. 5C). Juveniles
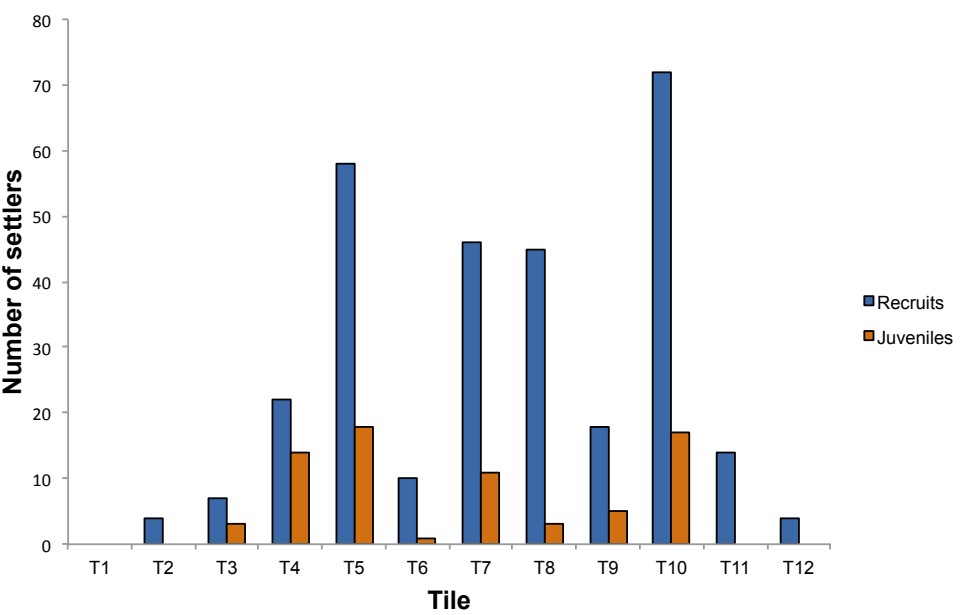

**Figure 3    Number of settlers on the tiles deployed in the Colombara cave.**

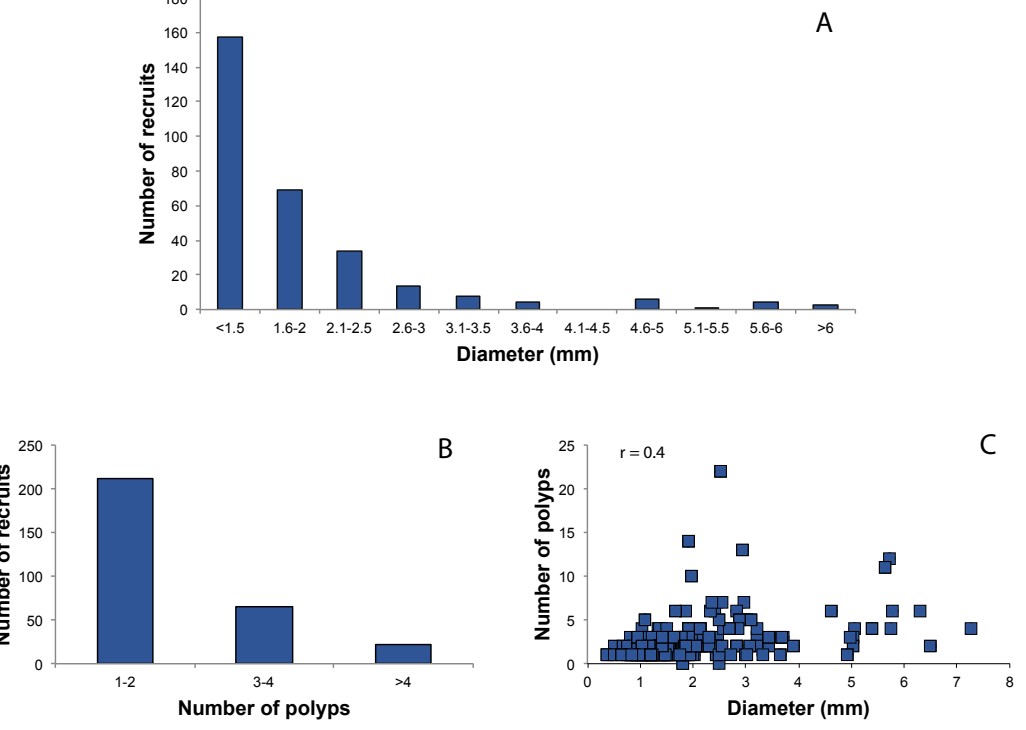

**Figure 4    Distribution of red coral recruits.** (A) Diameter classes, (B) number of polyp's classes, (C) relationship between number of polyps and diameter. r.: Pearson coefficient.

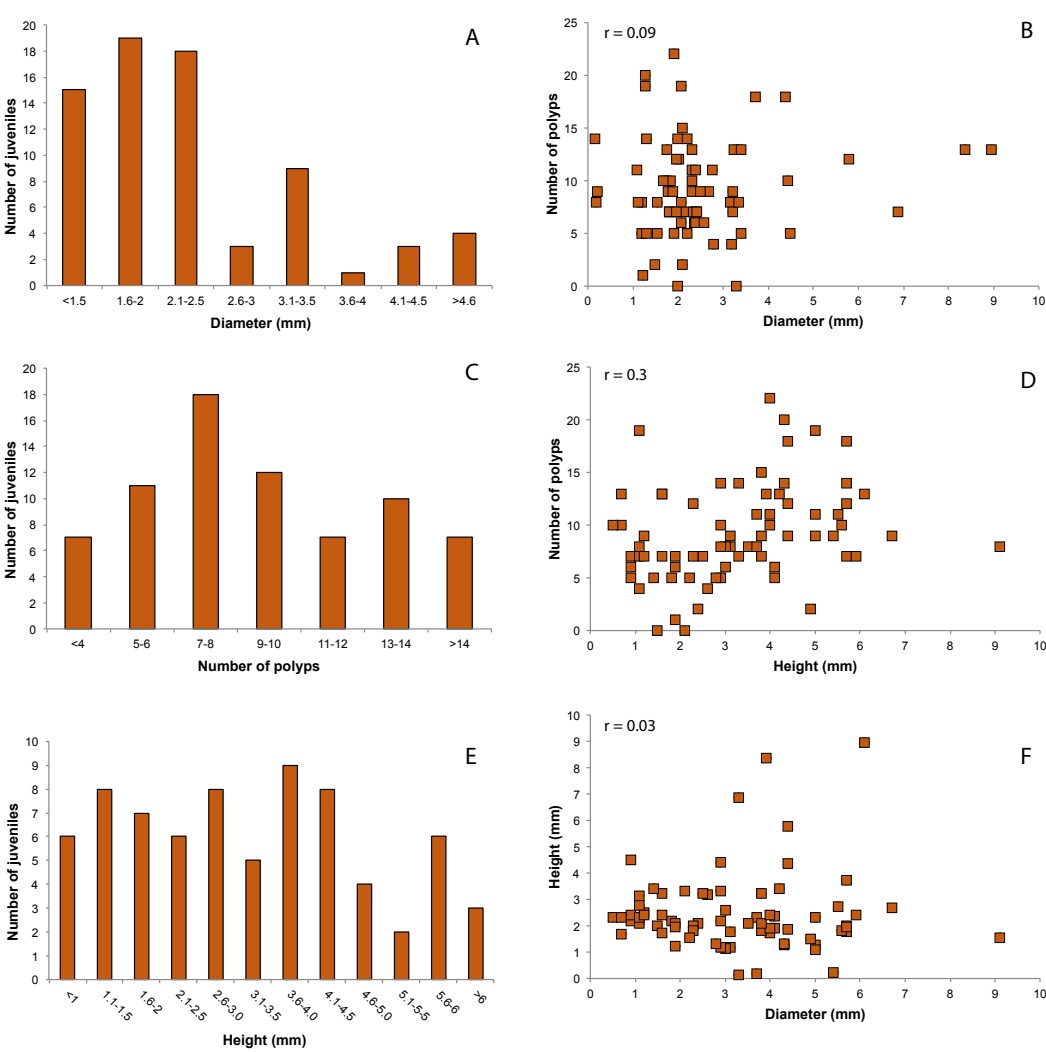

**Figure 5** **Distribution of red coral juveniles.** (A) Diameter classes, (C) number of polyps classes, (E) height classes. Relationship between: (B) number of polyps and diameter, (D) number of polyps and height, (F) height and diameter. r.: Pearson coefficient.

with 3–4 polyps were 9.7% of the total number, while 90.3% had more than five polyps. The height of the juveniles ranged from 0.5 to 9.15 mm, with a mean of $3.3 \pm 1.8$ mm (Fig. 5E). Pearson's correlation coefficients showed no correlation between diameter and number of polyps ($r = 0.09$, Fig. 5B), nor between diameter and height ($r = 0.03$, Fig. 5F). A weak correlation between height and number of polyps was observed ($r = 0.3$, Fig. 5D).

## Genetic variability

Individuals in which more then two loci did not amplify due to technical failures (e.g., low DNA quantity, no amplification of after re-amplification) were not included in the dataset. A total of 290 red coral individuals were genotyped. No genotypic disequilibrium was observed between loci (all $P > 0.05$ after FDR correction). No evidence of scoring errors

**Table 1** Locus characteristics for all the individuals: number of individuals genotyped per locus (N); number of alleles per locus (Na); null allele frequency (R); gene diversity ($H_e$, Nei 1973); observed heterozygosity ($H_o$); *Weir & Cockerham (1984)* estimator of $F_{IS}$ (*f*).

|  | N | Na | R | $H_e$ | $H_o$ | *f* |
|---|---|---|---|---|---|---|
| Cor9 | 286 | 23 | 0.22 | 0.81 | 0.31 | 0.61[*] |
| Mic20 | 289 | 10 | 0.1 | 0.6 | 0.64 | 0.21[*] |
| Mic24 | 283 | 18 | 0.09 | 0.67 | 0.43 | 0.37[*] |
| Mic26 | 283 | 23 | 0.25 | 0.6 | 0.13 | 0.79[*] |
| Cor46 | 218 | 19 | 0.19 | 0.29 | 0.04 | 0.86[*] |
| L7 | 283 | 20 | 0.1 | 0.56 | 0.32 | 0.44[*] |
| Cor58 | 235 | 16 | 0.25 | 0.66 | 0.17 | 0.75[*] |

**Notes.**
[*]Significant deviation from panmixia after false discovery rate correction at 0.05 (*Benjamini & Hochberg, 1995*).

due to stuttering or large allele dropout was found, according to MICROCHECKER. An excess of homozygotes was detected at all loci due to the presence of null alleles. Null allele frequencies ranged from 0.09 (Mic24) to 0.25 (Mic26 and Cor58). Number of alleles ranged from 10 (Mic20) to 23 (Cor9 and Mic26). The expected and observed heterozygosity ranged from 0.29 (Cor46) to 0.81(Cor9) and from 0.13 (Mic26) to 0.64 (Mic20), respectively. The estimators of $F_{IS}$ (*f*) were positive and ranged from 0.21 (Mic20) to 0.86 (Cor46) (Table 1).

Overall, a low genetic variability ($H_e = 0.59 \pm 0.16$ and $H_o = 0.29 \pm 0.20$; mean $\pm$ SD) was found. Out of the 290 individuals analysed, 281 different multilocus genotypes (MLGs) were identified. Four MLGs were found twice; one MLG was encountered three times and one four times. Individuals sharing the same MLG always came from the same tile. Tiles where identical MLGs were found were T7, T8 and T10. The individuals sharing the same MLG were between 0.2 cm and 13.65 cm apart (in T7 and T10, respectively). In T8 and T10, one recruit and one juvenile shared a MLG. The unbiased probability of identity ($P_{ID}$) that two sampled individuals share identical MLG by chance was $1.5e^{-04}$.

Juveniles and recruits did not show significant differences in terms of $H_e$ and $H_o$ ($P$ values associated with the permutation procedure: $P_{Ho} = 0.65$ and $P_{He} = 0.71$); while recruits showed a significantly higher allelic richness than juveniles ($P_{Ar} = 0.001$).

## Relatedness

A high degree of genetic relatedness among individuals was found. The mean observed relatedness was significantly different from the expected relatedness under panmixia (observed mean rxy = 0.027, resampled mean rxy = 0.014, $t = 1,173$, $P = 0.001$). Pairwise relatedness based on the *Queller & Goodnight (1989)* coefficient revealed that 25.98% of the pairwise comparisons were involved in one or more parentage relationships, with 8.57% and 17.41% of individuals involved in a full-sib and half-sib relationship, respectively. These percentages were numerically similar considering the relatedness within the two temporal cohorts separately.

The percentage of individuals correctly assigned at the same tile where they were found by the individual assignments method using GENECLASS was around 50%. The effective population size (Ne) was estimated as 68.7 (95% CI [42.5–116.4]) including all

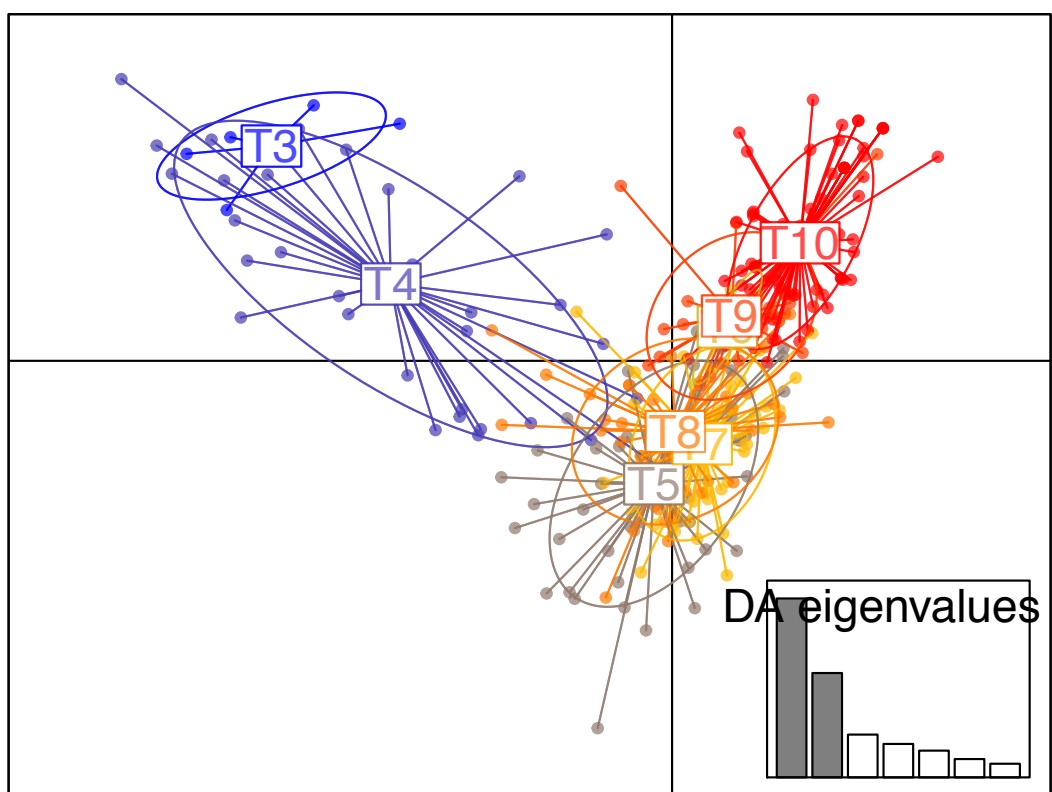

**Figure 6 Scatterplot of the discriminant analysis of principal components (DAPC) of the settlers found in the tiles deployed in ceiling of the Colombara cave.**

the individuals and as 32.5 for recruits (95% CI [20.6–57.8]) and −1,467.8 for juveniles (95% CI [87.1–∞]). The last negative value is expected when the population is sufficiently large that no notable linkage disequilibrium is induced through genetic drift (*Waples & Do, 2010*).

## Spatial and temporal genetic structure

No genetic structuring was observed between recruits and juveniles ($F_{ST} = 0.008$, $P = 0.16$). The clustering method identified $K = 2$ gene pools based on Evanno's delta K statistic (Figs. S1, S2) but with high levels of admixture as many individuals could not be undoubtedly assigned to either cluster. Nevertheless, an eastward genetic gradient of differentiation were observed. Regarding the results from the DAPC analysis, the plot of BIC as a function of the number of clusters $K$ (ranging from one to 16) did not present a real minimum (Fig. S3). Nevertheless, when using tiles as groups, the DAPC was in agreement with the Structure results (Fig. 6) with a higher genetic isolation of the titles T3 and T4 with all the other tiles.

Significantly positive kinship coefficients were detected between all the individuals within the single tile ($\phi$ij = 0.066, $p < 0.001$) indicating that individuals within tiles had a higher genetic relatedness than random pairs of individuals. Within the cave, the autocorrelogram suggested an estimated patch size of less than 40 cm (Fig. 7). The spatial autocorrelation

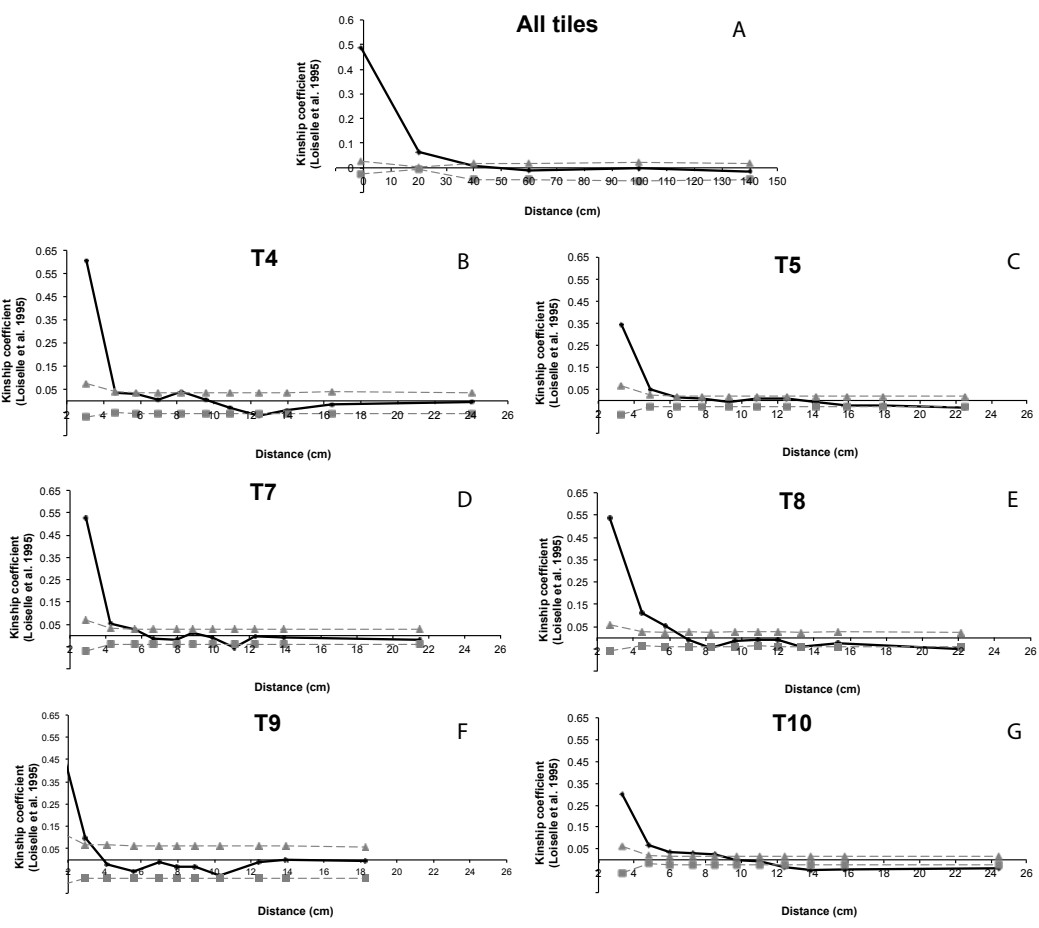

**Figure 7** **Spatial autocorrelation analysis of Loiselle kinship coefficient** (*Loiselle et al., 1995*). (A) All tiles: correlogram performed for all the individuals within the cave; (B–G) correlogram performed for for each tiles with enough numbers of settlers (T4, T5, T7, T8, T9 and T10). Grey lines represent 95% confidence intervals.

between individuals within each tile showed a significant positive value within a range from 3 to 10 cm.

## DISCUSSION

In the present study the early life history traits of the Mediterranean red coral have been investigated using settlement tiles deployed inside a submarine cave. Size and genetic structures of recruits have been analysed. Two cohorts of red coral recruits were collected and they showed significant variability in space and time. No significant genetic structure was observed between the two cohorts, while settlers on the same tiles were highly related, suggesting that larval clouds recruiting nearby are sibs.

In summers 2010 and 2011 *C. rubrum* successfully recruited on artificial tiles inside the cave. The density of settlers ($26.57 \pm 29.92$ individuals/400 cm$^2$) was higher compared to values found by *Bramanti et al. (2014)* in Cap de Creus marine reserve (Costa Brava,

Spain: 42°29.21′N; 03°30.18′E, Spain, 5.6 ± 2.8 individuals/400 cm$^2$) and in Portofino (Eastern Ligurian Sea, Italy: 44°18.18′N; 09°12.83′E, Italy, 17.5 ± 4.7 individuals/400 cm$^2$). While adult colonies dwells on the ceiling and on the walls of the cave, in the current study settlers' density significantly differed between these two habitats, with higher values on the tiles located on the ceiling of the cave. The only three tiles (T1, T13, and T14) where no recruitment was recorded were positioned on the walls of the cave. The causes of these differences are not fully understood yet, but they could be related to several factors. Both larval behaviour and habitat constraints could influence the higher recruitment density observed on the ceiling. Even if *C. rubrum* larvae could survive at least 16 days (potentially up to 42 days) in the plankton (*Martínez-Quintana et al., 2014*), inside caves and/or overhangs they probably are trapped by the ceiling due to their negative geotaxis (*Weinberg, 1979*; *Martínez-Quintana et al., 2014*). Sediment deposition is a limiting factor for red coral recruitment, and observations made by *Virgilio, Airoldi & Abbiati (2006)* showed that a thin coat of sediment covering vertical surfaces affects colony densities. These findings suggest that red coral populations dwelling in crevices and caves are more resilient due to enhanced recruitment rates, while populations living on cliffs and on rocky bottoms, which are the most currently exploited, might be endangered due to recruitment limitation.

Due to the difficulties in identifying tiny individuals (*Bramanti, Magagnini & Santangelo, 2003*), only a few studies have investigated the early life stage of this species (*Garrabou & Harmelin, 2002*; *Bramanti et al., 2005*; *Bramanti et al., 2007*). By using settlement tiles, it was possible to discriminate two early life stages of *C. rubrum*: recruits and juveniles. Significant variation in the abundance of recruits and juveniles were observed, suggesting both an inter-annual variability in larval supply and/or post-settlement mortality. However, it is not easy to disentangle these two processes, and this was not in the scope of this study. Concerning post-settlement processes, they include the intra- and inter-specific interactions mediated by chemical cues, food limitation, local water flow, predation and competition for space (*Fraschetti et al., 2002*). Competition for space, together with grazing intensity, is known to produce variations in coral recruitment at this spatial scale (*Babcock & Mundy, 1996*). All these processes, and their interactions, contribute to the high mortality rates in gorgonian recruits (*Caley et al., 1996*; *Perkol-Finkel et al., 2008*), including C. *rubrum* (*Garrabou & Harmelin, 2002*; *Bramanti, Magagnini & Santangelo, 2003*).

The number of polyps per individuals was consistent to *Bramanti et al. (2005)*, with a higher number of polyps in juveniles compared to recruits, and a significant correlation between number of polyps and height in the juveniles. Moreover, individuals settling on tiles showed a considerable variability in diameter and height, suggesting that growth rate in early red coral stages may be extremely variable and possibly much higher than the growth rates estimated from adult colonies (0.89 mm /year between the first and second years, *Bramanti et al., 2005*; but see Table S2 in *Cerrano et al., 2013*). Considering that in this study the age of colonies on tiles was 20 months (for the juveniles), an average annual growth rate in diameter of 1.48 mm /year for juveniles has been estimated. These results support the high variation in colony growth rates among geographic regions and in the early stages of colony life (*Santangelo et al., 2012*; *Cerrano et al., 2013*).
An important finding of the genetic characterization of settlers within the Colombara cave is that several identical multi locus genotypes (MLGs) were shared by recruits and juveniles. However, up to now, no evidence of asexual reproduction and/or polyp bail-out were reported in *Corallium rubrum*. The most likely explanation for the presence of identical MLGs is that, due to the low level of genetic variability of the species, these individuals are sibs.

The low levels of observed heterozygosity (mean Ho $= 0.29 \pm 0.20$) compared to that previously pointed out in natural populations (Mean Ho $= 0.32 \pm 0.04$, *Costantini, Fauvelot & Abbiati, 2007*; Ho $= 0.5$, *Ledoux et al., 2010a*), could be due to the small population size. However, this hypothesis seems unlikely considering the high density of settlers on the tiles, and the high average density of colonies on the Portofino Promontory ($227 \pm 37$ colonies $\times$ m$^2$; *Bavestrello et al., 2015*), including the Colombara cave (C Cerrano, pers. obs., 2010). The observed genetic variability suggests high genetic drift acting within the cave. In fact, a correlation between low genetic variability and low effective population size was already observed in *Corallium rubrum* (*Ledoux et al., 2015*) confirming that genetic drift might be relevant in this species (*Costantini, Fauvelot & Abbiati, 2007*; *Ledoux et al., 2010a*). Moreover, as tested by *Costantini, Fauvelot & Abbiati (2007)* the presence of spatial and/or temporal mixing of differentiated gene pools (e.g., Wahlund effect) could explain the previous observed high levels of heterozygosity.

The significant deviations from Hardy–Weinberg equilibrium, emphasized by the high positive $F_{IS}$ estimates, it is unlikely to be related to null alleles. Occurrence of null alleles in red coral populations has been previously reported (*Costantini, Fauvelot & Abbiati, 2007*), however, in this study their frequency is low and it should not affect observed heterozygosity.

High inbreeding rates (e.g., mating between relatives) are a more likely explanation for the $F_{IS}$ values. This phenomenon was already observed in *Corallium rubrum* (*Costantini, Fauvelot & Abbiati, 2007*). This hypothesis is also supported by the occurrence of sibs on a single tile, suggesting limited larval dispersal and/or collective dispersal (*Broquet & Yearsley, 2012*). Settlers' consanguinity could be explained by other factors (e.g., asynchrony of reproduction events, gametes' behaviour, uneven sex ratio and clonality). Little is known about red coral gametes behaviour (*Santangelo et al., 2003*) but other possible explanations seem unlikely. In fact, in *Corallium rubrum* the sex ratio is balanced (*Tsounis et al., 2006*; *Bramanti et al., 2014*; *Santangelo et al., 2015*), reproduction is not completely synchronous but occurs within a discrete time-interval in summer (*Santangelo et al., 2003*), and up to now clonality (asexual reproduction) was not observed. Moreover, no genetic structure was observed between the two analysed cohorts of settlers. All these findings, including the high relatedness among settlers (full and half-sib relationship), suggest that in both cohorts larval supply was provided by a limited number of genetically similar adult colonies (or progenitors). Fine spatial scale genetic structure is a common feature of gorgonians (*Brazeau, Sammarco & Atchison, 2011*), including red coral (*Costantini, Fauvelot & Abbiati, 2007*; *Ledoux et al., 2010a*). In the Colombara cave a population patch size of about 8 cm was detected, of the same range observed by *Ledoux et al. (2010a)* in a Mediterranean marine cave close to Marseilles (20–30 cm). These high SGS, together with the low genetic

variability and the high inbreeding rate, in a close environment as a marine cave, could enhance local adaptation. However, further reduction of genetic diversity, and hence reduction of population size due to, for example, global changes (e.g., thermal anomalies, acidification), could lead to overcoming the "inbreeding threshold", resulting in reductions of both fitness and of the risk of local extinction (*Frankham, 1995*).

## CONCLUSION

The present study provides new insight concerning recruitment processes in red coral populations using a submarine cave as an experimental mesocosm. The main outcome of the study is that *C. rubrum* individuals settling in the Colombara cave are highly related at very small spatial scales, and that most larvae recruiting nearby are sibs. Evidence of the processes explaining this pattern cannot be provided, however, self-recruitment and the presence of clouds of larvae that settle all together could be possible explanations. Parentage analyses between adult individuals, both inside and outside the cave, and the recruits would help to disentangle the two processes. Understanding processes acting in the early life history of a species is a challenging but crucial task, with major implications for conservation. In fact, these processes drive the population structure and dynamic of the species, and are essential for the resistance and resilience of populations.

## ACKNOWLEDGEMENTS

We would like to thank the director of the Portofino MPA Dr. Giorgio Fanciulli, and the Università Politecnica delle Marche for the authorization of the field experiments. We thank Adriana Villamor for her advice during the lab work; Massimo Ponti for his help in the imaging data analysis, Marco Palma and Ubaldo Pantaleo for field assistance and Paolo Mancuso for his great drawing capacity. Part of this research has been performed in the frame of the Mesomed project in collaboration with Portofino Divers. This work is part of the PRIN 2010–2011 (prot. 2010Z8HJ5M), and PRIN 2015 (prot. 2015J922E4) projects by the Italian Ministry of Education, University and Research.

### Funding

The authors received no funding for this work.

### Competing Interests

The authors declare there are no competing interests.

### Author Contributions

- Federica Costantini conceived and designed the experiments, analyzed the data, prepared figures and/or tables, authored or reviewed drafts of the paper, approved the final draft.
- Luca Rugiu analyzed the data, prepared figures and/or tables, authored or reviewed drafts of the paper, approved the final draft.

- Carlo Cerrano conceived and designed the experiments, performed the experiments, authored or reviewed drafts of the paper, approved the final draft.
- Marco Abbiati conceived and designed the experiments, contributed reagents/materials/analysis tools, authored or reviewed drafts of the paper, approved the final draft.

## Field Study Permissions

The following information was supplied relating to field study approvals (i.e., approving body and any reference numbers):

Field experiments were approved by the Director of the Portofino MPA and by the Università Politecnica delle Marche (Authorization n. 3/2011 (n. prot. 449/2-1-5.) and authorization no 4/2012 (Protoc. No 409/2-1-1)).

## Data Availability

The raw data are provided in a Supplemental File.

## Supplemental Information

Supplemental information for this article can be found online at http://dx.doi.org/10.7717/peerj.4649#supplemental-information.

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
