# Peer review of "Living upside down: patterns of red coral settlement in a cave"

_PeerJ, doi:10.7717/peerj.4649_

## Round 0.1 · original submission · Major Revisions

All three reviewers have given constructive comments. In particular, two of three (and I agree) stress that there could be more work put into defining and explaining your research question(s) and thinking, and then also linking your thoughts from paragraph to paragraph. Thus, while the scientific part of your work only needs minor revisions, the text will need some extensive reworking or answers to reviewers, and thus my decision is major revisions are needed.

·

Basic reporting

Costantini et al. present the results of an experimental study conducted on the field, examining the recruitment patterns of red coral Corallium rubrum on settlement tiles deployed in a Mediterranean submarine cave, using genetic approaches, to investigate the relatedness and genetic structuring within and among cohorts in order to provide additional information on early life characteristics and population dynamics in this species. This is a very interesting study that was surely not that easy to set up, and it provides results that are new in this field for this species (relatedness among recruits) Overall the manuscript is well written, (though I have made small comments and corrections on the annotated pdf), data analysis are sounded, though there are missing information which needs to be added (mostly in Material and Method section, see below). Two figures (1 and 2) need improvement (both in terms of quality and mission information). See general comments for details. There are several issues (described hereafter) in the manuscript that I think the authors can easily address.

Experimental design

The submission does not really define a research question. There are no statements regarding how the study contributes to filling a possible gap. I suggest the authors to rewrite the end of their discussion by adding why their study is important in the context of what we know and what we do not know in this species and more generally in sessile marine invertebrates. In other term: why did the authors specifically investigated the recruitment patterns and genetic relationships among red coral recruits within a marine cave? What were the specific questions to answer when setting up this their experimental design?

My second point is that information regarding the experimental design and material and method are partially missing to fully understand what has been done exactly. The first missing information concerns the adult population within the cave (Line 109 and after): what were the adult population densities within this cave? on the ceiling versus on the walls? The second missing information concerns the cave itself: The description and figure of the cave is not clear enough: if it stands between 29 and 39meters, it means that the opening of the cave is 10 meters, but how wide is the opening? how deep is the cave (from the opening to the end?) It seems that figure 1 provides a view of the cave from the opening. I strongly suggest the authors to provide as well a complete profile of the cave (example: on the left the opening with the ceiling at 29m depth and the bottom at 39m depth, and one the extreme right the end (final wall) of the cave. Also, the authors should not forget the scales on their maps and drawing. And the third and final missing information that is probably the most important one, is how exactly the tiles were placed inside the cave: how were the tiles fixed compared to adult colonies? in between? how far from the adult colonies? How were they placed from each other: randomly placed? or placed regularly following a strait transect? or differently? It is written line 215 that tiles were 20cm distant but if a tile is 20x20cm, and the distance between 2 tiles is 20cm, it means that this distance represent the distance between the edges of the tiles, not the distance between the center of each square... this should be clarified somewhere: how exactly were placed the tiles? And how far were they fixed from the opening of the cave? 1 meter after the opening, further deeper in the cave? were they differences in light between the tiles? please provide all this missing information.

My third point concerns the distinction made between recruits and juveniles cohorts. It is written line 132-135: “Based on the literature on red coral early life stages (Bramanti et al., 2005) the age of each individual was estimated on the basis of its height: individuals with an encrusting button shape were assigned to the cohort 2011 (hereafter recruits); individuals that developed in height forming a small branch were assigned to the cohort 2010 (hereafter juveniles).” In this cited article, Baramanti and coauthors identified cohorts “based on the identification provided by 4 year's worth of transparencies”, underwater pictures were taken every year and then compared to the tile once removed after four years. They also indicated “As measurements of settler height were extremely difficult due to their small size, only cohorts 2, 3 and 4 were examined for this parameter”.
As recognized by the authors, line 340-342, “individuals settling on tiles showed a considerable variability in diameter and height, suggesting that growth rate in early red coral stages may be extremely variable and possibly much higher than the growth rates estimated in adult colonies (Bramanti et al. 2005; but see Table S2 in Cerrano et al. (2013)).“. Indeed, in this present manuscript, from all measurements presented in the different figures, we can see that diameter and number of polyps are overlapping between recruits and juveniles: Therefore, how was exactly made the distinction between the two cohorts? what is the accuracy in height estimates for the juveniles? in other term, what is the confidence interval around each height estimate? how can you be sure that a 0.5 mm height individual is not a fast growing recruit rather than a slow growing juvenile? could you not just use a gap in a height distribution among all 372 individuals? or such gap does not exist?
Therefore, it seems to me that some additional information is missing here: was there a threshold in height between recruits and juveniles? were all colonies of height close to 0 considered recruits while juveniles had a minimum height of 0.5mm (based on figure 5e)? when mixing several measures (height, diameter, nb of polyps, max width), were each individuals attributed to one of the other cohorts? i.e. were the individual points organised in two clouds?

Validity of the findings

One of the most important issue I see in this manuscript is that the conclusions drawn from the results are overstated (Line 32-33 and line 367 and after). The authors show in their study high relatedness among individuals located on the same tiles (i.e. at very small spatial scale), suggesting that most larvae recruiting nearby are sibs and that their parental colonies are in their close vicinity (though this is not checked). Then the authors conclude that the cave population is likely closed and self-recruiting (“The results suggest that in the cave the C. rubrum population is closed and self-recruiting and that a small proportion of individuals contribute to the recruitment” in the resume and in the conclusion: “The Colombara cave seems to host a closed and self-recruiting population”). But showing high relatedness at very small spatial scales does not necessarily mean that the population is closed and self-recruiting. Indeed, one can imagine that this could be the result of a cloud of larvae, possibly coming from outside the cave, that have settle all together within the cave. I do not really believe in such hypothesis, but no one can reject it. If the authors want to demonstrate that the red coral population within the cave is closed and mostly self recruiting, they need to conduct parentage analysis between recruits and adult colonies to show that adults from the cave are actually the parents and only parents of the recruits. There may be indeed high self-recruitment rate within the cave, but the data presented here do not provide evidence for this. Even if it is true that the individuals analysed in the cave likely belong to a single homogeneous population (since you did not find genetic structuration among you individuals in the cave; see line 368) it does not necessarily imply a closed population restricted to the cave, unless the authors compared recruits and juveniles to other surrounding populations and found significant differences among them. Therefore I suggest strongly suggest the authors to rewritte their main conclusion based on what they results truly show and avoid over speculating.

Additional comments

Here are additional remarks for manuscript improvement:

Line 25 : provide name and location of the cave

Line 28: indicate the purpose of genotyping with microsatellite

Line 61: it is the first time that settlement tiles are cited, how can they be "also used"? remove also and adjust the sentence: example: "Settlement tiles are commonly used in coral populations to provide reliable measures of recruitment rates and variability, as well as individual growth rates...

Line 86-91: awkward sentence, please revise and divide in two: the red coral is one of the most abundant species (reference). then provides indication regarding its distribution (with references).


line 118-120: “Ten PVC tiles were also fixed in a vertical wall outside the cave and at the same depth (Faro Point) where the red coral population has an adult density of 425 ± 100 colonies x m2 (Bavestrello et al., 2015). In summer 2011 the tiles were removed a no a single recruit was found.”
This sentence is a little bit confusing as I think it is not well placed. Indeed, because it is written here that the tiles (without saying which ones) were removed in summer 2011 and that no recruits were found, it is confusing since here you are talking about the tiles located on the outside of the cave, not the one inside (as it stands, it is not clear). I therefore suggest placing these two sentences after the main experiment explanations because it is more of a note to the reader, therefore line 136, and rewrite it as: “it should be noted that 10 additional PVC tiles were fixed outside of the cave at the same time that the one inside the cave, on a vertical wall X meters away from the mouth of the cave where red population densities reach 425 colonies per m2, but that no recruits were found when checking the tiles during summer 2011.”

Line 122: if you remove the two previous sentences as suggested, you can then start this sentence with "In February 2012, after two reproductive events (summer 2010 and summer 2011), the tiles were removed (n=14 as 2 located on one vertical wall were lost, Fig. 1)".

Line 199: “the number of genetic clusters, K, was estimated…” estimated from what? Within each cohort or within the entire sample, i.e. recruits + juveniles?

Line 215, as mentioned previously (remark of line 116), the indication of how the tiles are placed is missing: every 20cm along a transect? if yes, you do not need to say that the distance between two tiles is 20cm, because this is only true for the closest tiles. You can though indicate that the distance between two tiles is estimated based on the center of each tile (and not the gap between two tiles)
This said, if a tile is 20x20cm, and the distance between 2 tiles is 20cm, it means that this distance represent the distance between the edges of the tiles, not the distance between the center of each square... this should be clarified somewhere: how exactly were placed the tiles?

Line 229 – 233: invert the two sentences: precise first how many individuals were found per categories, then where they were found

Line 237-239: can this not be checked genetically? If several planula merged into a recruit/colony, they should host several DNA: maybe the authors should provide the information on whether they have encountered or not multipic profiles for some individuals indicative of mixed DNA origins?

Line 246: See also my remark for line 132-135: this seems very very tiny... does it mean that all recruits had height smaller than 0.5mm? was there a height threshold to separate recruits from juveniles? 0.5? or all recruits were 0 in height?

Line 248-250: what is C.I? confidence interval? Pearson's correlation are rather given by giving Pearson's R value (or R2), and its associated p-value.

Line 249: “ a weak correlation”, replace by “a weak but significant correlation”

Line 253: why not all 372 individuals? it was by choice or because of technical failures? Please indicate.

Line 261-267: this is an important finding which deserve to be discussed: several identical MLG were found among the recruits and juveniles (i.e. likely clones considering the rather small chance of sharing the same MLG by chance: 1.5 10-4). The fact that they were mostly found on the exact same tile is an indication that it is not just by chance. Could you provide some possible explanations for this? Does anyone ever test for clones among colonies, and ever reported a similar finding? Could there be polyp bail-out?

Line 270: how did you estimate allelic richness? it is not indicated in the material and methods. Was the richness estimated for a common sample size? Please indicate

Line 343-351 : yes indeed but these references report estimates within natural populations, not within cohorts. Therefore you cannot directly compare your results to the results cited therein.
Indeed, an alternative hypothesis is that not all adults participate to the reproduction events, explaining why genetic diversity within cohorts is smaller than the overall population genetic diversity.

Figure 1 need improvement of the quality (but see also my remark regarding the scheme of the figure which needs to be more detailed: most importantly scales are missing)

Figure 2: scales are missing on each photograph

Reviewer 2 ·

Basic reporting

The article is clear, well written and organised. The references and scientific background are maybe limited (see attachment);

Experimental design

The results are original and the methods are adequate. The positioning of the study and the results in a more general context is less clear.

Validity of the findings

The data are robust and well analysed. The discussion is a little bit short and sometimes lacks precision.

Additional comments

see attached file

Annotated reviews are not available for download in order to protect the identity of reviewers who chose to remain anonymous.

·

Basic reporting

There are two main issues with the manuscript, valid for each section:

1) There is a general lack of obvious logical links between ideas, sentences and paragraphs. The reader spends more time trying to follow the authors’ train of thought than actually reading the manuscript. The authors really need to improve the fluidity of the text using linking markers.

2) The general scientific context is not clear. What scientific gap is filling this study? What are the important or main scientific questions answered by this study? The reader has the impression that a lot of work has been done, and well done, but the story is missing: what links all this work?

Specific comments
Line 38 – This statement is extremely general and needs bibliographic reference and examples. In addition, “sessile benthic organisms” would be more accurate.

Line38-49 – I would re-organize this paragraph to match the order of the first sentence, i.e. factors affecting dispersal, then factors settlement and then factors affecting recruitment. It would clarify the ideas and unify the paragraph. In addition, even though the authors cite pertinent examples, they don’t cite reviews on these topics and they do not present these examples as such.

Line 47-49 – I would weaken or modify the authors statement. Indeed, the authors present low dispersal and local recruitment as a threat to species survival. However, there is a large body of literature underlining that local recruitment could enhance populations survival through local adaptation (see Sandford and Kelly (2011) Local Adaptation in Marine Invertebrates. Annual Review of Marine Sciences 3: 509-535).

Line 52-56 – There is an incredibly large body of literature in these topics, especially marine invertebrates’ population genetics: the authors should include more references here, or at least reviews on these subjects.

Line 64 – This is badly worded: how would you differentiate local from external recruitment within a cohort without knowing the genetic structure of the populations in the study area? As C. rubrum larvae are planktonic, even if all individuals from a cohort are related doesn’t imply that they were produced locally: they could come from another population. Please consider rewording and developing that sentence and provide relevant bibliographic references.

Line 84-85 – The protection plans of the Italian government, even though important, is no use for this publication. Please delete the last part of this sentence or put it in the Acknowledgment section.

Line 86-92 – These sentences are quite confusing and are repeating themselves. How come the red coral is one of the most abundant species in shallow Mediterranean underwater caves if it has been harvested for hundreds of years? I would separate the ideas: one sentence describing the ecological niche of that species, and one sentence describing its current distribution and abundance.

Line 102-104 – There is clearly a room for improvement here as the point and objectives of this study seem unclear. Why using this species in particular? What are the problematic and hypotheses? What are the objectives? You could even include the expected outcomes.

Line 183-190 – The objective of this analysis is confusing. What is the reference tile of an individual?

Line 187 – There is a typo “…migrants among tiles._Red coral…”

Line 225 – Do not start a sentence with an abbreviated Latin name.

Line 245 – Please reword “…number of 9.23 ± 4.6. 9.7% of the…”, the succession of numbers is difficult to read.

Line 248 – Please replace “and” after the parenthesis by “, nor between”

Line 281-286 – As in the methods, the objective of this analysis is unclear and needs to be explained.

Figure 1 – Please increase the resolution of the figure. It is difficult to understand in which region of northern Italy this study was conduct. Please include a scale in order to appreciation the size of the cave.

Figure 2 – Please include a scale on each photo.

Figure 3 – The x-axis legend seems to be written with several text policies, please homogenize. Avoid the unnecessary use of abbreviation, unless they are clearly in the legend: either wright “N: number of settlers” at the end of your legend or wright the y-axis legend in whole letters.

Figure 4 – As for Figure 3, avoid unnecessary abbreviations or explained them.

Figure 5 – Same comment as before. Please considere re-ordering the figure such as A and B are on the same row, C and D on the same row etc.

Table 1 – Please cite “Benjamini and Hochberg (1995) Controlling the False Discovery Rate - a Practical and Powerful Approach to Multiple Testing. Journal of the Royal Statistical Society. Series B, Statistical methodology, 57, 289-300.” in the manuscript and in bibliographic references.

Experimental design

The methods used in this study, even though lacking some precisions, are pretty well founded and well explained, which would make them easy to replicate. All sampling and research permits are given by the authors, assuring high ethical standards. However, as stated before, it is not totally clear yet what knowledge gap is filled by the study: a clear statement of problematic, objectives and hypotheses are a requirement.

Specific comments
Line 112-114 – Valid sampling and research permit are extremely important but consider including them in the Acknowledgments section of the publication.

Line 116 – Why did the author use PVC tiles rather than clay or glass? Was there a technical reason or was it to compare the results with previous studies? A recent publication proved PVC to be an excellent recruitment substrat (Kennedy EV, Ordoñez A, Lewis BE, Diaz-Pulido G (2017) Comparison of recruitment tile materials for monitoring coralline algae responses to a changing climate. Mar Ecol Prog Ser 569:129-144.), but a rational of this choice could be a valid addition to the methods section.

Line 120-121 – Please precise (1) which tiles were removed and (2) how, were and when recruitment was estimated.

Line 126 – If relevant, please precise the version of the software.

Line 137-138 – Please precise which software was used to conduct the statistical analyses.

Line 152-153 – Did you use the same PCR protocol as in the references cited? Please precise.

Line 154 – Did you estimated the amount of linkage disequilibrium? Please correct accordingly.

Line 186-187 – Can red coral reproduce one year after settlement? What mechanism could explain first year migrants? Please develop on the reproductive biology of this species in the introduction to justify this analysis.

Line 191 – Why the authors didn’t use a method based in linkage disequilibrium (as Waples, and Do. 2008. LDNe: a program for estimating effective population size from data on linkage disequilibrium. Mol. Ecol. Res. 8:753–756.). It has been proven to be one of the most reliable Ne estimation methods in a wide variety of scenarios (Gilbert and Whitlock (2015) Evaluating methods for estimating local effective population size with and without migration. Evolution, 69, 2154-2166).

Line 197 – How many replicates did you use? Only one sub-dataset of randomly selected individuals will not give you reliable results.

Line 208 – Assignment analysis using Structure assume no linkage disequilibrium and Hardy-Weinberg equilibrium within populations, which is not the case here. Please study the clustering of the population using DAPC, which relies on less assumptions (Jombart T., Devillard S., Balloux F. 2010. Discriminant analysis of principal components: A new method for the analysis of genetically structured populations. BMC genetics 11:94.).

Line 236 – The authors only give one number here; what trend is decreasing? The wording is confusing, and a graph representing number of recruits according to size might be a valid addition here.

Line 253 – The authors said 372 individuals were sampled, but only 290 were genotyped. What happened to the missing individuals?

Validity of the findings

The conclusions of this study are all based on the results, and the authors did not make spurious conclusions and assumptions. However, the last step is often missing: how are the results included in a more general context? For example, the high mortality post settlement: how is that important for the species survival and conservation? Is it only a gorgonian trait or is it common in Cnidarians? The authors need to develop their discussion, notably by comparing their results with other species or organisms, which would increase the scope of their study.

Specific comments
Line 303-321 – The authors separate their conclusions and explanations from relevant bibliographic references supporting them, e.g. line 312“…related to larval behavior (Martínez-Quintana et al., 2014)…” and line 316“…they probably remain trapped in the ceiling due to their active upward swimming behavior…”. I would reorganize the paragraph in such a way that logical blocks are not separated (do not separate results explanations from bibliographic references supporting them) as it is done in the paragraph from line 322 to line 336

Line 352-359 – Other factors could explain the high consanguinity of this population: asynchrony of reproduction events, gametes behavior, uneven sex ratio, clonality…. Please develop this part of the discussion.

Line 359-360 – As stated in the introduction, local reproduction and recruitment can enhance local adaptation and speciation. Please develop this section to explain why, in this case, high inbreeding is a problem.

Line 375-376 – This sentence, notably “…that the two reproductive events interested individuals belong to the same genetic pool.” is difficult to understand. What are interested individuals?

Line 380-383 – Reading this sentence, it is unclear what brick was added to the wall. Indeed, this study does not clearly describe the effect of environmental factors (even though the effect of tile positions in the cave was tested), but rather the final outcome of all environmental factors and biological processes on the genetic diversity and structure of C. rubrum cohorts. Please rephrase this final conclusion to clarify what was found in this study and why is it relevant/important in the general context of the study of C. rubrum ecology.

Additional comments

In short, this study, which well designed and conducted, is lacking a clear general context. It is hidden at the moment: its proper statement, as the specific objectives of the study, will greatly improve the manuscript. The authors need to work on the fluidity of the text: all the parts for a good publication are there, they just need to be put together.

---

## Round 0.2 · Minor Revisions

Your manuscript is greatly improved! There are still some points to be ironed out, but I agree with the reviewers that no new analyses are needed, and thus my decision is minor revisions are needed. I look forward to seeing a revised version.

Reviewer 2 ·

Basic reporting

The authors made a good job at correcting this article and the introduction is now more clear. The objectives are well defined and the article is well written, even if some mistakes are present in the modified parts. The figures are clear, and the figure of the cave is really useful.

Experimental design

The research question is now well defined and most methods are well described (see my comments below for some specific points)

Validity of the findings

The findings are robuts to the data and most conclusions are well stated. Nevertheless there are some cases of incorrect use of previous publications (see below) leading to speculations.

Additional comments

Abstract :
- likely to « be » sibs
- « most larvae originated from adult colonies surrounding the tiles » do the results demonstrate this ? See the previous comments by other reviewers as well

Introduction :
line 48 : larval dispersal and recruitment do not always lead to chaotic genetic patchiness, rephrase
line 53 : why « nevertheless » ?
line 66 : estimates of what ? I don’t see how SGS could estimate recruitment rate
lines 67-68 : do you have references supporting the mention of strong SGS in corals apart from your model species ? And the references on Corallium do not demonstrate a philopatric behaviour.
Line 80 : rephrase : if a species lives outside a cave, this can’t act as a refugia ?
Lines 91-92 : as all species, this species is affected by stressors along its entire geographical area, but I guess none of the cited studies specifically tested the impact of a given stressor on the whole range ?
Lines 95-96 :what reference supports this statement ?
Line 99 : Padron & Guizien 2015 is not in the bibliography. And does it refer to a loss of expected heterozygosity (Hexp) or to a departure from panmixia (Hexp vs Hobs) ?

Methods :
Line 147 : « colonies with » height equal to zero...
Line 161 : indicate if the ANOVA test is based on permutations
Line 187 : if the N=54 refers to the threshold for rarefaction, this should come at the end of the sentence and be specified
Line 197 : are you sure of the rxy values ? See for example the manual of Identix : http://www.genetix.univ-montp2.fr/identix_ms.pdf
Lines 222-223 : explain the meaning of this resampling : is it to test differentiation with similar sample sizes between recruits and juveniles ? I guess that the number of replicates is simply too low to evaluate this effect. And the procedure implemented in GENEPOP should partly take this into account ?
Line 256 : delete « overall »
Line 262 : the number of significant digits seems too high
Line 274 : « correlation between these two variables »
Lines 279-282 : a correlation coefficient is not a test of correlation. Did you perform a test ?
Line 285 : « due to technical failure »
Lines 301-302 : was this probability PID identical for the different cases of repeated MLGs ?
Lines 308-309 : what do you compare with the t-test (i.e. what is your null hypothesis) ?
Lines 310-311 : give the p-values for the two tests, only one p-value is given here
Line 318 : « estimated in... » : this seems incorrect, rephrase
Line 328 : for the DAPC give in Supplementary Material the graph of the BIC according to the number of clusters
Lines 325-328 : is there a difference in cluster membership between the different classes of individuals ?
Line 358 : what is the information in the reference Cau et al. 2016 ?
Lines 379-382 : for the same notation for the units (and for years it’s a)
Lines 388-389 : when considering the number of alleles and the expected heterozygosities this is not really convincing. This species does not seem to display such low genetic diversity.
Lines 390-391 : give some numbers for comparison
Line 395 : a related explanation could be that other studies estimated diversity based on samples comprising different cohorts. Whereas considering only one or two cohorts might give lower diversities.
Line 405 : what larval behaviour ? Is there a demonstration of the impact of this behaviour on FIS ?
Line 416 : « or progenitors » ?
Lines 421-423 : the article of Bramanti et al. (2009) does not test inbreeding depression : remove this sentence or find an adequate reference (see comment on the previous version as well)
Line 426 : « the loss of fitness » ? and « reduction of fitness » would be more adequate

·

Basic reporting

This new version of “Living upside down: patterns of red coral settlement in a cave” is a giant leap forward, improving all previous flaws. The text is way more fluid (with just a few typos indicated in the attached pdf), easy to read, follow and understand. In addition, a huge work has been done on improving the bibliographic review, integrating relevant and pertinent references and giving this study a general context that was previously unclear. A similar attention has been given to figures: more precise, better organized and with clear legends. All the results given by the authors are relevant to fill the gap of knowledge they identified and stated in the introduction.
On a small downside, a last work on logical links between paragraphs of the introduction could be performed, but the manuscript is totally understandable as it is right now.

Experimental design

As I said in my first review, the analyses performed were almost all well explained and relevant. The authors improve their explanations when needed and included supplemental analyses when asked for (analyses that supported their results), with the same attention to details so that anyone could replicate the study. The only small detail that I would include is the Bayesian Information Criterion graph from the DAPC analyse as a supplementary figure: that would not change the results and there is no imperial need to comment it, but it is important.

Validity of the findings

Again, the authors made an incredible improvement on the discussion part. Their discussion is fluid, with good summarizations of their results and comparisons to relevant literature, but also identifying speculations and future possible work. Their data is easily usable and can be transferred to other formats quite easily. The lack of knowledge they identified in the introduction is not totally filled, but the authors are aware of this fact and this paper is good step to do so. Finally, the authors managed to introduce their research in a general context, a major flaw of the previous version of the manuscript.

Additional comments

The amount of work produced by the authors is impressive, and it pays. This manuscript is fluid, with clear hypotheses and conclusions. The re-organization of several paragraphs, in addition to the bibliographic work in the introduction and discussion section, makes also a huge difference.

---

## Round 0.3 · Minor Revisions

I have gone over your submitted version myself, and it is almost ready for acceptance. Still, there are a few small English (and other) areas that need some final clarification, which I have listed below. Please excuse these comments now; I prefer to do this once I have an almost final version of a manuscript, as earlier texts may change based on reviewers' comments.

Note all line numbers from tracked changes MSWord file unless otherwise noted.

1. line 23: Add "a" before "natural laboratory", or make plural.
2. line 32: Should "relationship" be plural?
3. line 52: Add a comma after "(Martínez-Quintana et al., 2014)"
4. line 53: Delete "The" before "settlement". Also line 55 as well.
5. You state on lines 58-59: "Larval dispersal and recruitment play a primary role in maintaining genetic diversity, but they can undergo to high levels of stochasticity which may contribute to a chaotic genetic patchiness".
I am not sure what you mean here - particularly the phrase "undergo to high levels of stochasticity" - can you explain this more clearly please?
6. line 90: Can you define clearly what you mean by "corals" here please? If needed, provide taxonomic information.
7. line 93: "population" to "population's".
8. line 99: What do you mean by "the dominant currents"? Do you mean perhaps just "strong currents"? Might need to add a qualifier such as "often" as well.
9. line 101: replace "they" with "caves".
10. line 108: "down-facing" to "downwards-facing" would be better.
11. lines 113-117 - this sentence seems a bit awkward, can you rewrite it please?
12. line 121: You use present perfect tense "has been" in the first half of the sentence, and then past "was" is the second half. Easiest to change "was" to "has been".
13. lines 143-44: Add directions to the Lat and Long please.
14. line 146: "northwest" seems better than "North-West".
15. lines 153-4: "a steel screw" to plural to match "tiles"; or "each was fixed..."
16. line 174: replace "upcoming" with "subsequent".
17. line 182: add "the" before "red coral population"; and change to "...density as that of the cave.."
18. line 183: Any reason your depths are listed as deep to shallow and not (more usual) the reverse?
19. line 189: Delete "the" before "Pearson's".
20. line 212: Change "corals" to "coral".
21. line 215: "share" to "shared".
22. line 244: Add a comma after "(Piry et al., 2004)".
23. line 246: "belongs" to "belonged".
24. line 255: delete comma after "Waples".
25. line 264: replace "were" with "was".
26. line 288: "proved" to "proven".
27. lines 291-93: "analyses" is plural yet the sentence is written with singular in mind; please edit.
28. line 308: "on the wall" is better.
29. line 310: "checking" to "checked".
30. line 315: "be forming" to "have been formed".
31. line 326: delete "the" before "9.7%". Delete "of them" as well.
32. line 353: "away" replaced with "apart".
33. line 371: "in 68.7" should be "as 68.7", also check line 372.
34. line 393 - please change "is" to "was" after "the DAPC".
35. line 405: "recruit" to plural.
36. line 411: "compared".
37. line 418: "vault" - do you mean ceiling here? Also, "settlers" to "settlers'". Finally, this sentence - you are referring to the current study? If so, please mention this, it is is a bit unclear here.
38. line 423: "can" to "could" as this is speculative. Change line 424 "could" to "can" as this is fact.
39. line 427: "observations". Delete comma after "surfaces" on the next line.
40. line 431: "nowadays" seems somewhat colloquial - can you rephrase?
41. line 434: "a few studies".
42. line 454 and area: Again, make this clear this is the current study. Also, "age" to "the age", and perhaps "is" to "was" - past tense.
43. line 461: "was" to "has been". Delete comma after "that" on line 462. "sharing all the genotypes" makes no sense.... what do you mean by this?
44. line 467, 479 - "heterozigosity" is usually "heterozygosity".
45. line 478: "high-observed heterozigosity" - you mean that "high levels of heterozygosity were observed"? Also, please add "levels of" on line 467 before "observed".
46. line 475: delete "the" before "genetic drift".
47. lines 480-81: This sentence reads like a fragment, please reword.
48. line 486: "a single tiles"?
49. line 490: "gametes' behaviour". Delete "also the" as well.
50. line 496: "synchrony" to "synchronous".
51. line 497: add comma after "(Santangelo et al., 2003)".
52. lines 509-510: please change this section to: "... could lead to overcoming the ‘inbreeding threshold,’ resulting in reductions of both fitness and of the risk of local extinction..."
53. lines 70, 517: "Evidences" should be singular, this is not countable.
54. line 519: "altogether" should be "all together"?
55. line 520: "analysis" should be plural.

---

## Round 0.4 · accepted · Accept

You have revised the manuscript very thoroughly, and this work is now ready to be published. I look forward to seeing the published version.

Some final notes; please ensure that:

1. the spelling of "threatened" on line 93,
2. add "and" before "heat waves" on line 95,
3. "permutation" as plural on line 167
are corrected by the proof stage or earlier.

#